# One-Step Rapid and Sensitive ASFV p30 Antibody Detection via Nanoplasmonic Biosensors

Ya Zhao,[a,b] Rui Li,[c] ChangJie Lv,[a,b] Yuanfeng Zhang,[a,d] Hanlin Zhou,[e] Xiaohan Xia,[a,b] Shiman Yu,[d] Yongqi Wang,[d] Liping Huang,[c,e] Qiang Zhang,[f] Gang L. Liu,[c] Meilin Jin[a,b]

aCollege of Veterinary Medicine, Huazhong Agricultural University, Wuhan, China
bNational Key Laboratory of Agricultural Microbiology, Huazhong Agricultural University, Wuhan, China
cCollege of Life Science and Technology, Huazhong University of Science and Technology, Wuhan, Peoples Republic of China
dResearch Institute of Wuhan Keqian biology Co., Ltd, Wuhan, China
eLiangzhun (Shanghai) Industrial Co. Ltd., Shanghai, China
fCollege of Biomedicine and Health, Huazhong Agricultural University, Wuhan, China

Ya Zhao and Rui Li contributed equally to this study. The order was determined by the corresponding author after negotiation.

**ABSTRACT**  African swine fever (ASF) is one of the most serious transnational swine diseases in the world. The case fatality rate of susceptible pigs is up to 100%. Currently, no commercial vaccine is available, so the prevention and control of ASF mainly relies on early diagnosis and culling of infected pigs. As the ASF virus continues to evolve, develop, and diversify, nucleic acid testing becomes less efficient. Here, we developed a method for the rapid and direct optical measurement of African swine fever virus (ASFV) antibody *in vitro*. This one-step procedure requires nearly no sample preparation and involves p30 protein-specific label-free integration into standard 96-well plates. Using a nanoplasmonic biosensor with extraordinary optical transmission (EOT) effect, one-step sample addition, ASFV antibody was detected within 20 min. The positive antibody showed a satisfactory sensitivity and linear relationship in the dilution ratio of 1:100–1:16000. It was used for the detection of clinical serum samples with a coincidence rate of 96.6%. The measurement results can be automatically analyzed and displayed on a conventional microplate meter computer and connected device. Our detection method can be widely applied in point-of-care testing (POCT) of ASFV antibody in pig farms.

**IMPORTANCE**  African swine fever (ASF) is a serious transnational disease caused by the African swine fever virus (ASFV), which is highly contagious in wild boars and domestic pigs. There is currently no available vaccine for ASF; therefore, development efforts are a key priority as ASFV continues to evolve and diversify. The ASF antibody rapid detection platform comprising the nanoplasmonic biosensor with extraordinary optical transmission effect can greatly reduce the detection time and improve detection flux while maintaining detection sensitivity and specificity. The one-step sample addition can effectively avoid cross contamination of samples in the detection process. The detection method provides a solution for the rapid and accurate real-time monitoring of ASF in pig farms.

**KEYWORDS**  ASFV, p30 protein, one-step sample addition, antibody detection, nanoplasmonic biosensor

African swine fever (ASF), caused by the African swine fever virus (ASFV), is a highly contagious disease affecting wild boars and domestic pigs. Clinically, it is mainly characterized as an acute, febrile, hemorrhagic, high morbidity, and high mortality contact disease. The World Animal Health Organization (OIE) has listed ASF as a legally reported animal disease. ASF seriously threatens the international trade of pig-related industries and pork-related food safety and has caused huge economic losses to the global pig industry (1, 2). It was

Address correspondence to Liping Huang, lphuang@aliyun.com, Qiang Zhang, zhangq_0401@mail.hzau.edu.cn, Gang L. Liu, loganliu@hust.edu.cn, or Meilin Jin, jml8328@126.com.

The authors declare no conflict of interest.

first reported in Kenya, Africa in 1921, and then spread across the 3 continents of Asia, Europe, and Africa. It was introduced to China in August 2018, and after widespread circulation, has become the largest threat to China's swine industry (3, 4). ASFV has a multi-layered coating structure and is a large double-stranded regular icosahedron DNA virus. The outer capsid protects the inner protein and nucleic acid, which allows the virus to exhibit strong environmental tolerance (5). Wild boars and *Ornithodoros moubata* (African hut tampan [tick]) have become the natural storage hosts of the ASFV (6). The long-term existence and spread of ASF in wild boars and domestic pigs contributes to the diverse genetic variation of the virus. Currently, at least 24 genotypes have been identified based on the B646L gene (P72). Genotype II is mainly prevalent in China, for which strain variants record a low fatality rate. Genotype I, which only causes chronic infection, has also been isolated and reported (7, 8). Currently, the lack of a vaccine and effective treatment against ASFV and the complex genetic variation of the virus are leading to an increase in viral incubation period, decrease in viral load, and increase in the proportion of attenuated strains in asymptomatic or only mildly symptomatic pigs infected by attenuated and non-haemadsorbing (non-HAD) strains. These factors contribute to the reduced efficiency of nucleic acid detection in pigs (8). However, the ASFV antibody remains a clear feature of infection (9). Therefore, the development of a fast, sensitive, stable, and highly automated ASFV antibody detection method, alone or in combination with nucleic acid detection, is particularly important for the early diagnosis and control of ASF in the field of rapid detection in pig farms.

So far, owing to their high stability and sensitivity, enzyme-linked immunosorbent assay (ELISA) and indirect immunoperoxidase technique (IPT) tests are the most widely used laboratory immunology-based diagnostic methods in EU reference laboratories and OIE (10). New material-based techniques include duplex lateral flow, multiple fluorescence-labeled-antigen, and Luciferase Immunoprecipitation System assays, designed for rapid detection of ASFV antibodies (11–13). However, the procedures for IPT and ELISA are complex and time-consuming, and the advanced new material-based technologies cannot meet the stability and high-throughput of large-scale detection requirements. Therefore, a serological diagnostic method for ASF with easy-operation, high-automation, low time consumption, and suitability for high-throughput screening is urgently required to meet early clinical surveillance needs.

To satisfy the requirement of the above-mentioned ASFV antibody diagnostic technology for livestock husbandry, we proposed the nanoplasmonic biosensor detection platform, including the nanoplasmonic chip and microplate reader immunoassay analyzer. Surface Plasmon Resonance (SPR) is a method for detecting target molecules based on specific immune recognition and antigen binding. It is widely used in the detection of biomarkers and the identification and characterization of microbial and human immune responses (14–18). Specifically, Goat anti-pig or other host IgG was fixed on the chip surface, and the remaining sites were blocked with BSA. After the initial OD value was detected, serum samples and antigen labeled with gold particles were added in the wells in a one-step way, forming the Goat anti-host IgG-detect antibody-antigen sandwich complexes, the microplate reader captures the refractive change due to the complex at the bottom of the nanoplasmonic biosensors and displays it on the compute. Compared to ELISA, the nanoplasmonic biosensor has certain advantages including small sample volume requirement, rapid results, and increased sensitivity (19). In contrast to colloidal gold test strips, the nanoplasmonic biosensor platform has higher throughput and demonstrates greater practicability and consistency of test results (20). The principle of the nanoplasmonic biosensor platform is based on the effective combination of the p30 antigen protein labeled with gold nanoparticles and the corresponding antibody in the serum to achieve high sensitivity detection of virus antibodies. The nanoplasmonic biosensor platform can be used to quickly and easily detect ASFV antibodies, with high sensitivity, and has the potential for point-of-care testing (POCT) in pig farms. It is expected to be used for the diagnosis of ASF and ASFV antibodies in laboratories and farm monitoring.

## RESULTS

**Characterization of p30 protein and a nanoplasmonic resonance biosensor.** To prepare the recombinant antigenic protein p30 with high antigenic activity, we chose the

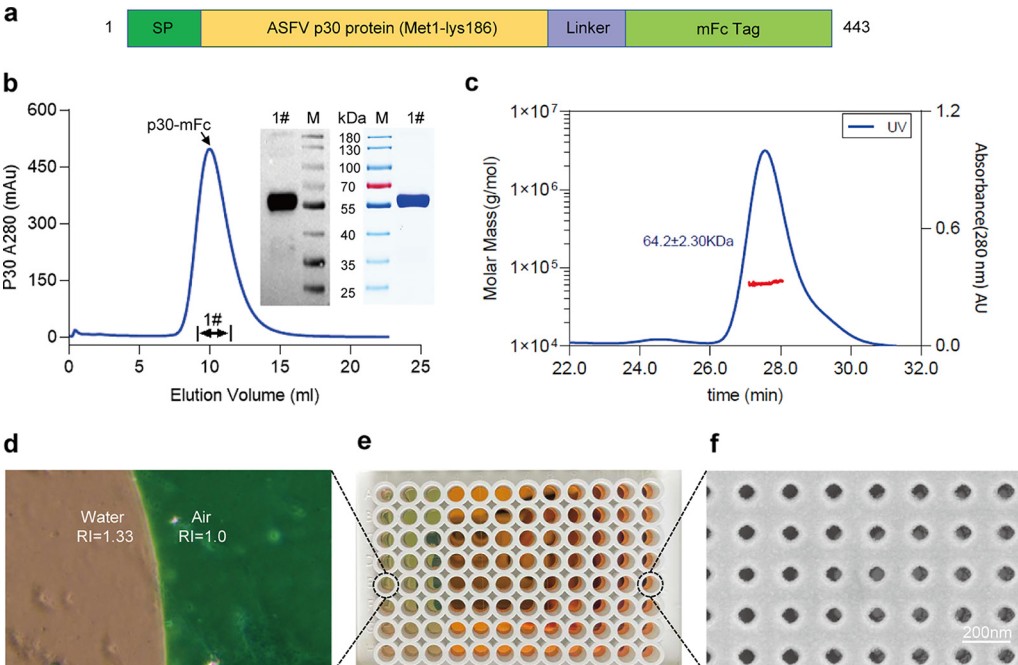

**FIG 1** Characterization of recombinant p30 protein and nanoplasmonic biosensor chip. (a) Schematic design of the p30 protein. (b) Representative elution chromatograph of the recombinant p30 protein on protein A column. Inset figures show SDS-PAGE and Western blotting analyses of the eluted samples. (c) The molecular weights of p30 protein were determined by SEC-MALS. (d) Boundary line between air and water in a hole with a microscope water and air show different colors (green and pink, respectively). (e) Integration of 96-well plate. The 3 columns on the left are filled with water to observe green color. (f) Scanning electron microscopy (SEM) image of replicated nanocup array.

CHO eukaryotic expression system. Recombinant p30 was designed as shown in Fig. 1a. A signal peptide sequence was constructed at the N-terminal to ensure protein secretion efficiency, while a mouse-IgG1Fc tag was linked to the C-terminal. Recombinant p30-mFc was successfully expressed from the cell culture supernatant and purified to high homogeneity. p30-mFc migrated as an approximately 55–70 kDa band under reducing conditions, and strongly reacted with anti-p30 rabbit polyclonal antibody in Western blot experiment (Fig. 1b). Moreover, SEC-MALS analysis indicated that the p30-mFc dimer was 64.2 $\pm$ 2.30 kDa and had purity greater than 98.5% (Fig. 1c). The nanoplasmonic resonance biosensor chip is an integration of a 96-well plate (Fig. 1e). Its surface presents distinct colors in different refractive index (RI) media, with pink and green in air (RI = 1.0) and water (RI = 1.33), respectively (Fig. 1d). Under scanning electron microscope, a high uniformity nanocup array structure can be captured (Fig. 1f). This indicates that the sensor chip has high sensitivity and homogeneity with a large response to spectral change of media alterations.

**Optimization of the nanoplasmonic assay conditions.** To maximize the detection performance of the nanoplasmonic biosensor, concentration of the coated antibody, type of serum sample diluent, concentration of AuNPs-p30 protein antigen, reaction time, and procedure were optimized. Figure 2a shows that the antibody is coated on the chip board coupled with MPA, and different antibodies have different effects on the relative OD response; 0.02 $\mu$g/mL Goat anti-pig IgG was the best response diluted concentration. To optimize the experiment more accurately and reasonably, and improve the signal-to-noise ratio, the following optimization experiment needs to, not only calculate the OD value of the standard positive serum, but also consider the positive sample minus the relative OD value (P-N value) of the negative sample, which is the maximum response value. There are many impurities in the serum samples, and different sample diluents also have a great impact on the test results. In order to obtain maximum value of P-N and reduce nonspecific reactions, and according to the screening results shown in Fig. 2b, the PBST buffer salt solution was chosen as the base dilution liquid. As shown in Fig. 2c, the maximum

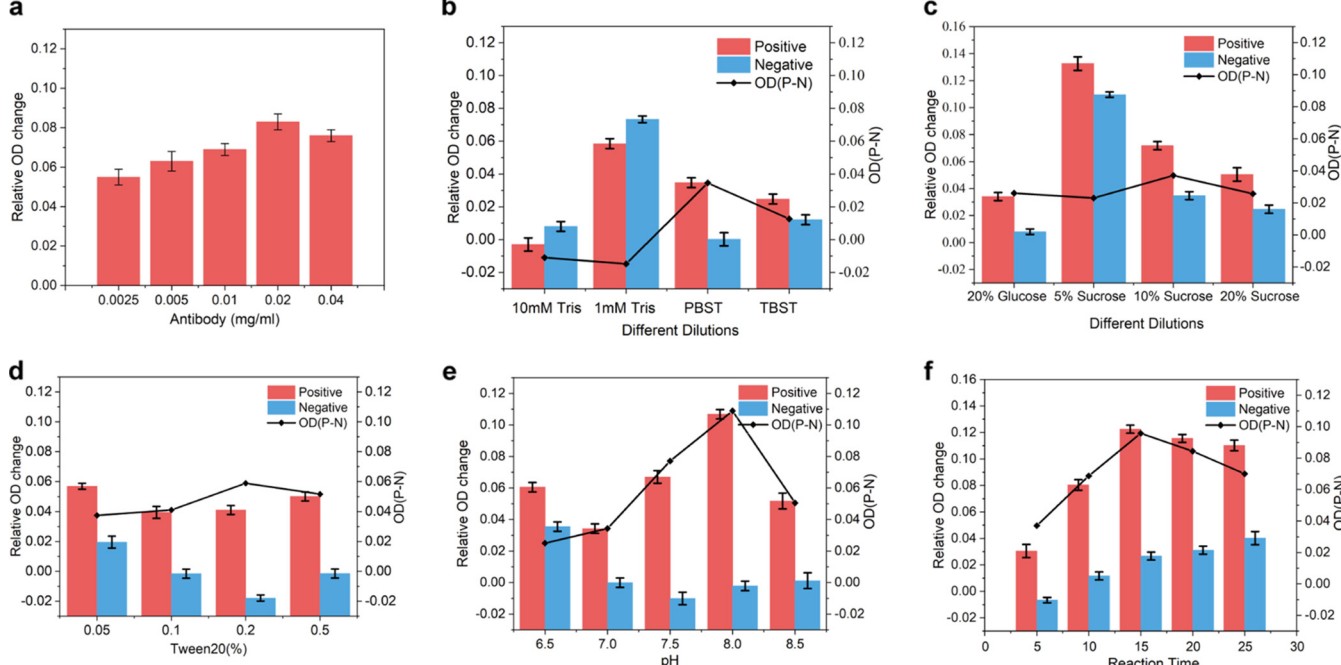

**FIG 2** Optimization of reaction conditions for nanoplasmonic biosensor assay. (a) Goat-Anti-pig IgG immobilization; concentration = 0.02 mg/mL; relative OD change reached the maximum. (b) PBST buffer salt solution as the base dilution liquid. (c) PBST base buffer was supplemented with 10% sucrose to improve the relative OD change. (d) The nonspecific reaction was reduced by adding surfactant 0.2% Tween 20 to PBST base buffer containing 10% sucrose. (e) At pH = 8.0, PBST diluent buffer containing 10% sucrose and 0.2% Tween 20 achieved maximum relative response. (f) At room temperature(24℃), AuNPs-p30, ASFV p30 antibodies, and goat anti-pig complex exhibited the maximum response OD value after shaking the hybrid for 15 min. Data are shown as means ± SD. Error bars represent the standard deviations from triplicates.

response is achieved when the base buffer contains 10% glucose to get the maximum response. On this basis, the surfactant (0.2% Tween 20) was added to decrease nonspecific reactions (Fig. 2d).

Next, in the optimization process for serum detection of ASFV, the pH value of the protein coupling process affected the coupling efficiency, as shown in Fig. 2e. This was done until the optimum stability of the conjugates was at pH 8.0, and the maximum response OD value was reached. Moreover, a PBST base salt solution containing 10% sucrose and 0.2% Tween 20 at pH 8.0 was identified as the best diluent used in the nanoplasmonic biosensor method. This was determined by using 3 dilution concentrations of the 3 standard positive serum samples (result not shown). Moreover, after balancing the relationship among detection range, detection sensitivity, and low background OD value, the optimal samples serum dilution ratio of 1:100 was confirmed.

Optimization of pre-reaction conditions before performing the detection was also implemented. Fifteen microliters of AuNPs-p30 (1.34 mg/mL) compound were pre-mixed with 1:100 diluted serum samples, and then immediately added to the bottom of the chip for shaking incubation reaction. During the reaction, AuNPs-p30, anti-p30 antibodies in serum and Goat anti-pig antibodies were combined to form a complex by continuous mild shaking at room temperature. The maximum response OD value was reached after shaking the hybrid for 15 min (Fig. 2f).

**Nanoplasmonic biosensor analytical performance.** A total of 246 sera samples were used to detect performance of the proposed method in practical applications. The cutoff value was verified by the ASFV antibody detection system based on the ROC curve. In addition, the same clinical samples were tested by Ingezim ELISA kit to verify the compliance rate of the detection methods (Table S1). As shown in Fig. 3a, when the relative OD value (cutoff value) is defined as -0.00085, the negative and positive serum can be clearly distinguished with good cluster. At the same time, ROC curve results showed that sensitivity and specificity were achieved compared with ELISA method (Fig. 3b). In addition, under suitable

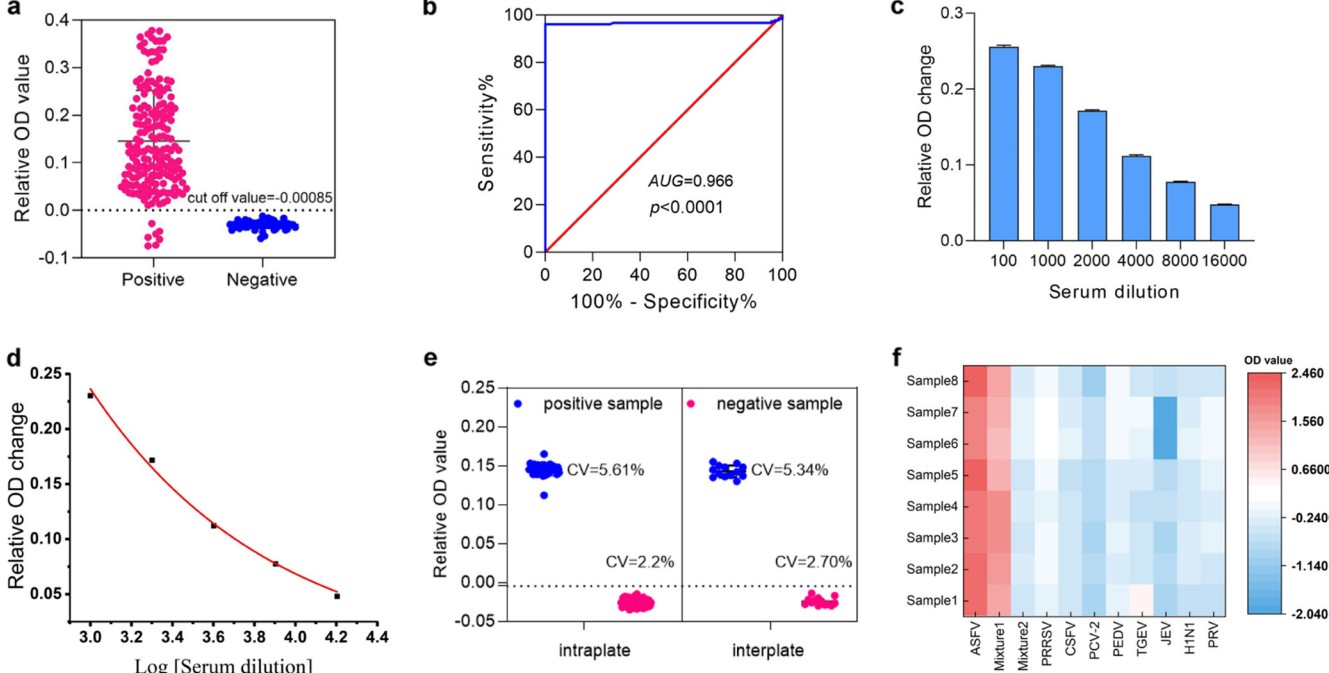

**FIG 3** Nanoplasmonic sensor analytical performances. (a) S-N relative OD change was obtained by the nanoplasmonic biosensor assay from a panel of 246 serum samples that were classified as positive or negative using the ELISA kit. (b) ROC curve based on the obtained data. (c) Sensitivity comparison between the nanoplasmonic biosensor and gold-standard indirect immunofluorescence for ASFV antibody detection. Three ASFV-positive sera were diluted from 1:100 to 1:16,000 fold. (d) Calibration curve based on the nanoplasmonic biosensor detection result of 3 ASFV-positive sera. (e) 48 times repeat test of positive and negative sera on nanoplasmonic biosensor, with 35 intraplate replicates and 13 interplate replicates, respectively. (f) Detection selectivity tested using different standard pathogens sera. Mixture 1 group included ASFV, PRRSV, CSFV, PCV-2, PEDV, TGEV, JEV, H1N1, and PRV. Mixture 2 group included PRRSV, CSFV, PCV-2, PEDV, TGEV, JEV, H1N1, and PRV.

experimental conditions, it is the most reliable method to determine the sensitivity of the detection procedure based on the known concentration of ASFV antibody. Since there was no standard concentration for ASFV-positive serum sample in our lab, we used the nano-plasmonic biosensor s to detect the dilution (1:100–1:16000) of 3 standard positive sera with unknown antibody concentration. In Fig. 3c and Fig. S1, even within a short reaction time of 20 min, the detection line is clearly distinguished when the serum dilution ratio is 1:16000; thus, nanoplasmonic biosensor sensitivity was consistent with that of the gold-standard indirect immunofluorescence. The calibration curve of the nanoplasmonic biosensor sensor was constructed by plotting the fit of various dilutions of the standard positive serum (1:100–1:16000). Figure 3d shows the calibration curve exhibiting a satisfactory linear range from 1:1000–1:16000 with a reliable correlation coefficient ($R^2 = 0.99$). The 48 times repeat test results of the positive and negative sample showed that the nanoplasmonic biosensor demonstrated good stability. Moreover, the coefficient of variation (CV) of the nanoplas-monic biosensor assay was less than 6% for intraplate assay and 3% for the interplate assay (Fig. 3e). CV value was much lower than the required 15%, indicating excellent per-formance and repeatability.

The specificity of the nanoplasmonic biosensor was evaluated by 100-fold dilution of the standard serum of common pig virus pathogens. As shown in Fig. 3f and Fig. S2 to S12, the relative OD values of the serum positive for these common pig pathogens were close to those of negative serum, but far from those of ASFV-positive serum, revealing successful differentiation. In addition, the relative OD value of the mixture 1 group (ASFV, PRRSV, CSFV, PCV-2, PEDV, TGEV, JEV, H1N1, and PRV) was much higher and closer to the ASFV-positive serum, while that of the mixture 2 group (PRRSV, CSFV, PCV-2, PEDV, TGEV, JEV, H1N1, and PRV) was much lower than the cut-off value. This implies that the proposed nanoplasmonic biosensor has no cross-reactivity with the standard serum positive for pig common virus pathogen, and shows good specificity and anti-interference ability in practical applications.

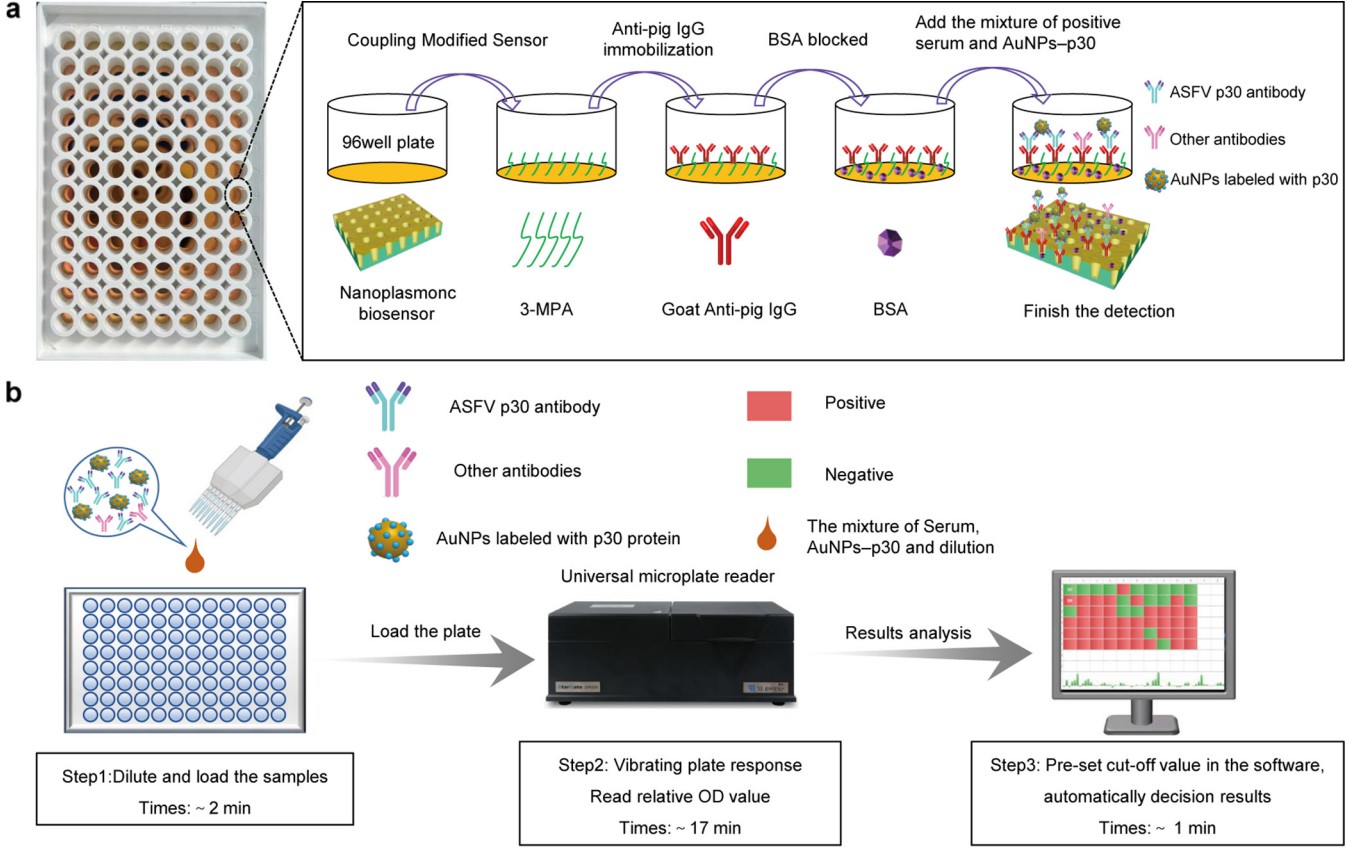

**FIG 4** Schemes of the nanoplasmonic biosensor assay for ASFV p30 antibody detection principle and process. (a) Schematic representation of nanoplasmonic biosensor chip surface functionalization as well as its ability to capture and detect ASFV p30 antibody. (b) Procedure for the determination of ASFV p30 antibody using the nanoplasmonic resonance biosensor.

## DISCUSSION

This study established a one-step detection method for ASFV antibody in pig serum, which has unique advantages compared with other antibody detection methods. First, the approach is based on an unlabeled nanoplasmonic biosensor with extraordinary optical transmission (EOT) effect integrated into a standard 96-well plate instead of a traditional ELISA plate (21, 22). The biosensor has the ability to produce a highly sensitive response to minute local refractive index changes. It can significantly shorten the detection time and amplify the detection sensitivity (19). Therefore, the biosensor is very sensitive to signal recognition. The nanoplasmonic biosensor method showed comparable sensitivity with indirect immunofluorescence in the positive samples dilution range of 1:100–1:16000; it was used for the detection of clinical serum samples with a coincidence rate of 96.6%. Compared with traditional ELISA, the required antigens and antibodies are not labeled in this method, so there is a wider selection spectrum of antagonists and antibodies. In addition, the reaction results are not affected by the enzyme catalytic process and enzyme reaction time. As shown in Fig. 4b, the method also has the advantages of one-step sample addition, automatic analysis, and fast generation of results. The computer and application-controlled microplate reader can output stable detection results within the total reaction time of 20 min; the detection time is shortened to the greatest extent. With this method compared to traditional ELISA, laborious processes including long incubation, comparative analysis, and repeated washing are simplified or replaced; when handling a large number of samples, such as in the case of liquid transfer, clappers steps, such as those for pollution possibilities, were effectively reduced. Therefore, this detection method overcomes the labor-intensity and demand for professional operating ability. In the practical application process, this can overcome the limitation of experimental conditions and space to a maximum extent, and detect a large number of serum samples in the shortest time.

Secondly, many studies have identified the p30 protein as an effective choice for the development of antigens for early detection of ASFV antibodies (23, 24). Therefore, the CP204L gene encoding ASFV p30 protein was cloned into the expression system of *Escherichia coli* or baculovirus for production of recombinant proteins (25, 26). In this study, the eukaryotic CHO expression system was used for the first time to obtain a high purity recombinant protein fused with mouse IgG1 antibody FC segment of p30 protein. Although there is no clear study on whether post-translational modification of p30 protein affects its immune activity, in indirect ELISA experiments, we found that compared with the p30 protein expressed by *E. coli*, the background value generated by serum nonspecific interfering components was lower and more conducive to eliminating the occurrence of a false positive rate (results not shown). Therefore, the coincidence rate between this method and commercial ELISA kit for detecting 246 serum samples was as high as 96.6%. The false-negative results could be attributed to the inconsistency of the antigen used in the 2 kits and the different times of antibody production (the p30 antibody was reported to be produced earlier than VP72). In addition, the fusion expression of mouse FC and p30 protein was innovatively carried out in this study, which was conducive to efficient expression and purification while ensuring stability of the recombinant protein. The p30 protein was coupled with gold nanoparticles (Au-NPs), enabling it to be preserved for several months at 2–4°C. Accelerated destruction tests also confirmed that the AuNP-coupled protein was more stable than the recombinant protein (results not shown). Previous studies have reported that this coupling with gold nanoparticles can significantly improve the detection sensitivity of nanoplasmonic biosensors (27, 28).

At present, the rapid detection nanoplasmonic biosensor platform also has some minor limitations. Since it shortens the incubation and washing times of antibodies to the maximum extent, this method has a higher requirement for fresh serum samples. Studies have reported that blood is clearly the main route of ASFV transmission; thus, the platform needs to be compatible with more sample types such as saliva and tissue fluid in order to reduce the transmission of disease caused by sample collection (29). In our previous study, we combined this chip with SARS-CoV-2 spike protein antibody or ACE2 receptor protein to develop a method for rapid detection of virions in oropharyngeal swabs of patients with early COVID-19 infection, whose minimum detection line was 370 vp/mL (19). Thus, it was clear that this rapid detection platform has strong plasticity and can help establish different antigen or antibody monitoring methods based on the characteristics of the objects to be tested.

In conclusion, based on the nanoplasmonic biosensor platform and eukaryotic p30 antigen, we developed an ASFV antibody detection device with characteristics such as short time consumption, high sensitivity, and simple operation, providing an efficient tool for early diagnosis and epidemiological monitoring of ASFV. Compared with the traditional commercial ELISA kit, the proposed method has excellent time efficiency (within 20 min), and enables high-throughput detection. In the next step, we plan to replace the connection between computer and microplate reader by a mobile phone, which will better improve the portability and application range of the device for future point-of-care testing (POCT) of ASFV antibodies in pig farms.

## MATERIALS AND METHODS

**Experimental materials.** A total of 246 pig serum samples were obtained and stored at −80°C until use. Of these, 186 clinical serum samples were collected from pig farms (Table S1); further, ASFV standard positive serum ($n = 20$) and specific pathogen-free (SPF) pig serum ($n = 40$) served as positive and negative controls, respectively, and were purchased from China Institute of Veterinary Drugs Control. Positive standard serum against other pathogens ($n = 8$), such as pseudorabies virus (PRV), porcine reproductive and respiratory syndrome virus (PRRSV), Japanese encephalitis virus (JEV), classical swine fever virus (CSFV), porcine circovirus type 2 (PCV-2), swine influenza virus (H1N1), porcine epidemic diarrhea virus (PEDV), and the transmissible gastroenteritis virus (TGEV), were preserved in our laboratory. A plasmid with mouse-IgG1-Fc gene, pCMV-mFc, was constructed in our laboratory. A CHO cell line was obtained from adherent cells and preserved in the laboratory. PrimerSTAR Max DNA polymerase, DNA Ladder, and competent cell preparation kit were purchased from TaKaRa. ClonExpress II One Step seamless Cloning Kit was purchased from Vazyme. Horseradish peroxidase (HRP)-conjugated affinipure Goat anti-swine IgG (H+L) was purchased from Proteintech. From Sigma-Aldrich, 3-mercaptopropionic acid (MPA), 1-ethyl-3-(dimethylaminopropyl) carbodiimide (EDC), N-hydroxysuccinimide (NHS), bovine serum albumin (BSA), ethanolamine, and phosphate-buffered saline (PBS) buffer were purchased.

Goat anti-Pig IgG (catalog no. V90401) was purchased from Solarbio Life Sciences Co., Ltd. AKTA Pure, Hitrap MabSelect PrismA unit, HyCell TransFX-C Medium, and Nitrocellulose (NC) membranes were obtained from Cytiva. The ASFV ELISA P72 enzyme-linked immunosorbent antibody detection kit was purchased from Ingenasa. All other chemical reagents used in the experiments were purchased from Sinopharm Chemical ReagentCo.

**Production and purification of recombinant fusion p30 protein.** The nucleotide sequence corresponding to the full-length sequence of ASFV-p30 with a signal peptide was synthesized by Tsingke Biotechnology Co., Ltd., and cloned into a pCMV-mFc expression vector. Recombinant plasmid pCMV-p30-mFc was transfected into CHO cells grown in serum-free medium. Stable and high yield cell clones were screened, and a fed-batch serum-free cell culture process in a 15-L bioreactor was developed. The supernatant was filtered and collected, and the protein was batch-purified by AKTA Pure 25 (sample load and elusion flow crate was 120 mL/h) at 4℃ with Prism Protein A resin (1 mL of sedimented resin/100 mL of supernatant). The collected elusion was pH adjusted to neutral with 1 M Tris-HCl (pH = 9.0) to prevent loss of antigen sensitivity caused by exposure to low pH. Finally, after 10 kDa ultrafiltration tube dialysis and 0.22 membrane filter steps, the recombinant fusion p30-mFc protein was obtained. According to the manufacturer's recommendations, protein concentrations were determined using a BCA protein assay kit (Sigma-Aldrich). The specificity and purity of the p30-mFc protein with anti-p30 rabbit polyclonal antibody (prepared in our lab) were examined using Western blotting and SDS-PAGE.

**Size exclusion chromatography coupled with multiangle light scattering.** Purified p30-mFc protein was analyzed by size exclusion chromatography coupled with multiangle light scattering (SEC-MALS). Samples (0.5 mL at 3.8 mg/mL) were loaded onto a Superdex 200 10/300 GL column (Cytiva; 0.5 mL/min in PBS gel filtration buffer) and passed through an Effective Optical System (EOS), Wyatt Technology Dawn Heleos II 18-angle laser photometer coupled to a Wyatt Optilab rEX refractive index detector (Wyatt Technology Corp.). Data were analyzed using the Astra 6 software (Wyatt Technology Corp.).

**Procedure for nanoplasmonic biosensor assay.** The principle of the immunological moieties in the nanoplasmonic biosensor assay is illustrated in Fig. 4a. The bottom of 3-MPA pre-modified nanoplasmonic biosensor was coated with goat anti-pig secondary antibody, and after BSA blocking, 50 $\mu$L water was added to read the full spectrum of the chip well. Then, the detected serum and AuNPs-p30 dilution were added. Antigen-antibody complexes were formed among sheep anti-pig secondary antibody, ASFV p30 antibody, and gold particle-labeled p30 protein. The antibody was sandwiched to the surface of the chip, resulting in the enhancement of the surface plasmon resonance (SPR) phenomenon, and the full spectrum OD value of the endpoint was read at this moment. The steps involved are as follows: first, the UV-curable polymer was evenly spread on the mold and placed on the polyethylene terephthalate (PET) sheet to produce the polymer nanocup array structure and then deposited on an array of 15 nm titanium and 60 nm gold. The sheet was cut to 14 cm × 10 cm and adhered to a 96-well plate with an opening at the bottom or a chip mold produced by a 3D printer (Object 30 prime; Stratasys Ltd.). Second, the chip surface was functionally modified, and the chip was incubated in a 10 mM MPA aqueous solution for 2 h at room temperature. The chip was cleaned twice with DI water and then immersed in a mixture of 400 mM EDC and 100 mM NHS for 30 min to activate it at room temperature. Next, the chip was rinsed with DI water and immediately fixed with Goat anti-pig IgG antibody at 20.0 $\mu$g/mL (in CBS), before overnight incubation in the refrigerator. The chip was rinsed with PBST, then incubated in 30 $\mu$g/mL BSA blocking solution for 30 min, and gently rinsed twice with DI water. After this step, the chip was ventilated and dried at room temperature and then stored in a humid environment at 4℃.

Finally, the p30 protein labeled with gold nanoparticles (AuNPs) was prepared. The gold nanometer was labeled, and the pH value of 1.5 mL AuNPs solution was adjusted to 8 using 0.1 M K2CO3 solution. Then, 8 $\mu$L 1.34 mg/mL p30 protein solution was added to the AuNPs solution and incubated for 15 min. After blocking with 22.5 $\mu$L blocking solution (10 mM PBS) containing PEG at 2 W (10%, wt/vol) for 15 min, the AuNPs solution was centrifuged at 7,500 rpm for 22 min. The precipitate of AuNPs-labeled p30 protein was resuspended in 275 $\mu$L stabilization buffer (10 mM Tris [pH 7.3], 0.3% sucrose and 0.05% PEG 20000) and stored at 4℃ for further experiments.

**Procedure of nanoplasmonic biosensor platform design.** Figure 4b shows the detection process of an automated system with sample addition, high-throughput readings, and real-time results display. First, the detection reagents are added to the biosensor wells of a 96-well plate. The plate is then loaded into a Tecan instrument (XLEMENT Shanghai Industrial Co., Ltd.) for a series of steps including automated start and end position readings, shaking, and analysis. Up to 96 samples can be processed and tested simultaneously. The sample to be tested is diluted 100 times using diluent R1, and 50 $\mu$L sample is added to the nanoplasmonic biosensor. Simultaneously, 15 $\mu$L of the p30 protein labeled with AuNPs is added to the chip. In principle, it depends on the antibody or protein in the serum of the test substance, as only the antibody in the ASF-infected serum can bind to the p30 protein. Tecan reads the starting point, and then shakes the nanoplasmonic biosensor to react for 15 min before reading the endpoint. SPR 100 software obtains the relative reaction OD value of the sample and determines test results (positive samples display red, negative samples display green).

**Indirect immunofluorescence assay.** Primary protospacer-adjacent motifs (PAMs) were seeded in 48-well plates and infected with ASFV at 0.1 multiplicity of infection (MOI) for 48 h. Next, cells were fixed with 4% paraformaldehyde and permeabilized with 0.1% Triton X-100. The cells were blocked with 1% BSA for 1 h at 37℃, and then incubated with the positive standard serum (1:100) against ASFV and other pathogens at 37℃ for 2 h. After washing three times with PBS for 5 min each time, the cells were incubated with CoraLite488-conjugated Goat anti-pig IgG (H+L) (Solarbio) at 37℃ for 1 h. After repeating the washing step with PBS, the samples were treated with DAPI for 10 min for nucleic acid staining. Samples were visualized with the EVOS FL Auto system (Thermo Fisher Scientific).

**Ethics statement.** All animal experiments and materials were reviewed, approved, and supervised by the Institutional Animal Care and Use Committee at the Huazhong Agricultural University (ID Number:

202208220001), and were performed in accordance with the Guidelines for the Care and Use of Laboratory Animals of the Research Ethics Committee, Huazhong Agricultural University, Hubei, China.

**Laboratory facility.** All cell experiments involving ASFV live viruses were performed in biosafety level 3 (BSL3) facilities in the Huazhong Agricultural University.

## SUPPLEMENTAL MATERIAL

Supplemental material is available online only.

**SUPPLEMENTAL FILE 1**, PDF file, 2.5 MB.

## ACKNOWLEDGMENTS

Y.Z., R.L., M.J., G.L., Q.Z., and L.H. conceived and designed the study. Y.Z. and R.L. wrote the report, drew the figures, and generated, analyzed, and interpreted data. C.L. collected the samples. Y.Z. and S.Y. conceptualized the antigen design and constructed a stable expression cell line. Y.Z., L.R., H.Z., C.L., X.X., Y.W., and Y.Z. performed the experiments. All authors contributed to manuscript editing and provided constructive feedback & final approval.

We report that no conflicts of interest are associated with this work.

This work was supported by the National Key R&D Program of China grant (No: 2021YFD1801405) and the National Nature Science Foundation of China (No: 91959107).

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
