## [Reviewer comments · Microbiology Spectrum]

Microbiology Spectrum

One-step rapid and sensitive ASFV p30 antibody detection via Nanoplasmonic biosensors

Ya Zhao, Rui Li, Changjie Lv, Yuanfeng Zhang, Hanlin Zhou, Xiaohan Xia, Shiman Yu, Yongqi Wang, Liping Huang, Qiang Zhang, Gang L Liu, and Meilin Jin

Corresponding Author(s): Meilin Jin, State Key Laboratory of Agricultural Microbiology, College of Veterinary Medicine, Huazhong Agricultural University

Review Timeline:

Submission Date:	June 23, 2022
Editorial Decision:	July 24, 2022
Revision Received:	September 23, 2022
Accepted:	September 24, 2022

Editor: Biao He

Reviewer(s): Disclosure of reviewer identity is with reference to reviewer comments included in decision letter(s). The following individuals involved in review of your submission have agreed to reveal their identity: Hua-Ji Qiu (Reviewer #1)

Transaction Report:

DOI: <https://doi.org/10.1128/spectrum.02343-22>

July 24, 2022

Prof. Meilin Jin
State Key Laboratory of Agricultural Microbiology, College of Veterinary Medicine, Huazhong Agricultural University
Wuhan
China

Re: Spectrum02343-22 (One-step rapid and sensitive ASFV P30 antibody detection via Nanoplasmonic sensors)

Dear Prof. Meilin Jin:

Thank you for submitting your manuscript to Microbiology Spectrum. Based on the comments of two experts, my decision at this time is "Modification". When submitting the revised version of your paper, please provide (1) point-by-point responses to the issues raised by the reviewers as file type "Response to Reviewers," not in your cover letter, and (2) a PDF file that indicates the changes from the original submission (by highlighting or underlining the changes) as file type "Marked Up Manuscript - For Review Only". Please use this link to submit your revised manuscript - we strongly recommend that you submit your paper within the next 60 days or reach out to me. Detailed instructions on submitting your revised paper are below.

Link Not Available

Sincerely,

Biao He

Journals Department
Reviewer comments:

Reviewer #1 (Comments for the Author):

Comments on VIRMET-D-22-00220

The present study by Ya Zhao et al. described a nanoplasmonic sensor-based method for the detection of antibodies against the p30 protein of African swine fever virus (ASFV). The work is novel and potentially useful. However, several concerns should be addressed.

Major concerns:

1. The cutoff value should be determined using more negative sera against ASFV.
2. The developed method should be evaluated using more sera (strongly/weakly positive or negative) samples with a clear background.
3. The claimed "one-step, rapid, and sensitive" detection should be justified and discussed.
4. The manuscript was poorly written and should be revised by native English speakers.

Reviewer #2 (Comments for the Author):

In the article titled "One-step rapid and sensitive ASFV P30 antibody detection via Nanoplasmonic sensors" the authors have described the development and validation of new point of care tool for the diagnosis of ASF. However, few points needs to be addressed related to the development and validation of the assay for the improvisation of the manuscript.

- 1.Line no. 60, The authors mention the low efficiency in detection of nucleic acid. Provide the justification and citation for the same.
- 2.The introduction section should described about the detection system of antibodies through nanoplasmonic biosensors.
3. Line no. 98. It is not clear whether these 164 pig samples were positive or suspected ASF specimens?
4. Line no. 99-100. The SPF free serum samples must have been used as negative control. Mention the same for better understanding of the reader.
5. Whether the animal ethical approvals for the study in experimental materials was taken?
6. Several other proteins such as p54, p72, p22 have shown to induce antibody response in pigs. Why specifically p30 was chosen to develop the assay?
- 7.Line no. 116-117. The reference standard test utilized in this study was ASFV p72 ELISA. However, the assay developed is based on p30 which is not appropriate to make comparison of sensitivity of the assay.
8. Line no. 141. How the mold and chip has been formed and made into 96 well plate format is not clearly explained.
- 9.Line no. 164. What is the detection reagent used for the assay?
- 10.Line no.162-175. The principle of binding of the immunological moieties is not clearly explained, even the detection system is not effectively written i.e, chromogenic, luminex.
- 11.Line 224-255. Authors have not explained the limit of detection of the assay, sensitivity and specificity with reference to gold standard.

Staff Comments:

Preparing Revision Guidelines

Please return the manuscript within 60 days; if you cannot complete the modification within this time period, please contact me. If you do not wish to modify the manuscript and prefer to submit it to another journal, please notify me of your decision immediately so that the manuscript may be formally withdrawn from consideration by Microbiology Spectrum.

September 20, 2022

Dear Editors and Reviewers,

Thank you for your letter and for giving us the opportunity to revise our manuscript entitled “**One-step rapid and sensitive ASFV p30 antibody detection via Nanoplasmonic biosensors**” (Spectrum02343-22). Those comments and suggestions are all valuable and helpful in revising and improving our manuscript, as well as providing an important guidance to our research. We have studied the comments carefully and have made corrections which we hope will meet with approval. The revisions are marked in blue in the revised manuscript. Additionally, these changes will not influence the content and framework of the paper. We have modified the manuscript accordingly, and the detailed corrections are listed point by point as follows:

The comments of reviewer 1 and corresponding responses.

Comment # 1: The cut-off value should be determined using more negative sera against ASFV.

Response: We are so grateful for your suggestion. Accordingly, in order to further determine the cut-off values, we added 20 negative serum samples on Line no. 105, the cut-off value and ROC curve in figure4a and Figure4b has also been modified accordingly.

Comment # 2: The developed method should be evaluated using more sera (strongly/weakly positive or negative) samples with a clear background.

Response: Thank you for your comment. As you say, in order to further evaluate the developed method, on the basis of 166 clinical sera samples, we added 20 clinical sera samples. The strength (strongly/weakly positive or negative) of the 186 clinical sera was assessed using the OIE-recommended ASFV VP72 Antibody ELISA kit (INGEZIM 11.PPA.K3; Spain), the blocking rate represents the strength of serum (blocking rate%, $X\% \geq 50\%$, ASFV positive; $X\% \leq 40\%$, ASFV negative; $50\% \geq X\% \geq 40\%$, Suspected samples). Among them, there are 47 sera samples with the blocking rate higher than 90%, 68 sera samples with the blocking rate between 60%-80%, 31 sera samples with the blocking rate between 50%-60%, and 40 sera samples with the blocking rate lower than 40%. Details are provided in Supplemental Table S1.

Comment # 3: The claimed "one-step, rapid, and sensitive" detection should be justified and discussed.

Response: Thank you for this valuable suggestion. We have suitably incorporated this in the discussion section according to your request on Line no. 295-305. Regarding the "one-step, rapid, and sensitive" advantages of the Nanoplasmonic biosensor compared to the traditional ELISA method, the Nanoplasmonic biosensor method showed comparable sensitivity with indirect immunofluorescence in the dilution range of 1:100–1:16000 Figure S1, it was used for the detection of clinical serum samples with a coincidence rate of 96.6 %. In addition, as shown in Figure 1b, the method has the advantages of one-step sample addition, automatic analysis, and fast generation of results. For further improvement, a computer and application-controlled microplate reader can be developed, so it can output stable detection results within the total reaction time of 20 min; the detection time can thereby be shortened to the greatest extent.

Comment # 4: The manuscript was poorly written and should be revised by native English speakers.

Response: Thank you for pointing this out. We apologize for the poor language of our manuscript. We worked on the manuscript over a long period of time, and the repeated addition and deletion of sentences and sections have evidently led to poor readability. We have now worked on both language and readability and also recruited native English speakers for language corrections. We sincerely hope that the flows of ideas and language clarity have been substantially improved.

The comments of reviewer 2 and corresponding responses.

Comment # 1: Line no. 60, The authors mention the low efficiency in detection of nucleic acid. Provide the justification and citation for the same.

Response: Thank you for raising this question. According to your suggestion, we have cited reference (8) in support of this statement.

Comment # 2: The introduction section should described about the detection system of antibodies through nanoplasmonic biosensors.

Response: We highly appreciate the reviewer for this insightful suggestion. The following text has been added in the Line no. 85-90: Specifically, Goat anti-pig or other host IgG was fixed on the chip surface, and the remaining sites were blocked with BSA. After the initial OD value was detected, serum samples and antigen labeled with gold particles were added in the wells in a one-step way, forming the Goat anti-host IgG-detect antibody-antigen sandwich complexes, the microplate reader captures the refractive change due to the complex at the bottom of the Nanoplasmonic biosensors and displays it on the compute.

Comment # 3: Line no. 98. It is not clear whether these 164 pig samples were positive or suspected ASF specimens?

Response: Thank you for your question. We are deeply sorry for the misunderstanding caused by the previous incorrect description. I have revised line no. 103-105. These sera are all clinically suspected samples.

Comment # 4: Line no. 99-100. The SPF free serum samples must have been used as negative control. Mention the same for better understanding of the reader.

Response: We are so grateful for this question. As you say, In fact, we are using SPF serum as the negative control, but due to the misunderstanding caused by the wrong description, we has been revised in Lines 103-106.

Comment # 5: Whether the animal ethical approvals for the study in experimental materials was taken?

Response: We are so grateful to you for raising this important question. All animal experiments materials were reviewed, approved, and supervised by the Institutional Animal Care and Use Committee at the Huazhong Agricultural University (ID Number: 202208220001), and were performed in accordance with the Guidelines for the Care and Use of Laboratory Animals of the Research Ethics Committee, Huazhong Agricultural University, Hubei, China. We have included this information in the revised manuscript.

Comment # 6: Several other proteins such as p54, p72, p22 have shown to induce antibody response in pigs. Why specifically p30 was chosen to develop the assay?

Response: Thank you for your careful review and this insightful question. As you say, p54, p72, p22, and other antigens also induce response in pigs. The p30 protein was used as the primary target in our study because it is highly immunogenic and evokes rapid immune response for ASFV early infection in pigs. This is critical for early diagnosis. We mentioned the rationale behind choosing p30 in Lines 316-319.

Comment # 7: Line no. 116-117. The reference standard test utilized in this study was ASFV p72 ELISA. However, the assay developed is based on p30 which is not appropriate to make comparison of sensitivity of the assay.

Response: Thank you for this question. Based on your comments, we also believe that it inappropriate to use two different methods to compare sensitivity. In order to further verify the sensitivity of our method, we added the gold-standard indirect immunofluorescence assay as the reference method, as detailed in Supplementary Figure S1.

Comment # 8: Line no. 141. How the mold and chip has been formed and made into 96 well plate format is not clearly explained.

Response: Thank you for this question. Nanoplasmonic sensor chips are manufactured using replication molding technology with molds. The original mold was a tapered

nanopillar array (diameter = 180 nm, depth = 450 nm, period = 400 nm) on a silicon wafer made using photolithography and plasma etching. The UV-curable polymer was evenly spread on the mold and placed on a polyethylene terephthalate (PET) sheet to produce the polymeric Nanocup array structure, after which 15 nm of titanium and 60 nm of gold were subsequently deposited on the polymeric Nanocup array in an electron beam evaporator. The sheet was cut to 14 cm × 10 cm and adhered to a 96-well plate with an opening at the bottom or a chip mold produced by a 3D printer (Object 30 prime Stratasys Ltd. USA).

Comment # 9: Line no. 164. What is the detection reagent used for the assay?

Response: Thank you for this question. The Nanoplasmonic biosensor platform relies primarily on plasmon resonance sensors with extraordinary light transport (EOT) effects for detection. The principle is based on the effective combination of the ASFV p30 antigen protein labeled with gold nanoparticles and the corresponding antibody in the serum to achieve high-sensitivity detection of ASFV p30 antibodies, which do not require the amplification step of enzyme-linked immunity. Therefore, the detection process only requires PBST buffer containing 10% sucrose and 0.2% Tween 20 (pH = 8.0) as serum diluents.

Comment # 10: Line no.162-175. The principle of binding of the immunological moieties is not clearly explained, even the detection system is not effectively written i.e, chromogenic, luminex.

Response: Thank you for this question. The principle of the immunological moieties in the Nanoplasmonic biosensor assay is as follows. The bottom of 3-MPA pre-modified Nanoplasmonic biosensor was coated with goat anti-pig secondary antibody, and after BSA blocking, 50 μ L of water was added to read the full spectrum of the chip well. Then, the detected serum and AuNPs-p30 dilution were added. Antigen-antibody complexes were formed among sheep anti-pig secondary antibody, ASFV p30 antibody, and gold particle-labeled p30 protein. The antibody was sandwiched to the surface of the biosensor, resulting in the enhancement of the Surface plasmon resonance (SPR) phenomenon, and the full spectrum OD value of the end point was read at this moment. The displacement difference before and after the detection sample is captured by the Universal microplate reader and read at the special absorbance of 580 nm and 610 nm. Therefore, the system converts the displacement difference generated by antigen antibody complex on the biosensor surface into signal, this detection system is different from chromogenic and luminex. We have added this section to Lines 150-158.

Comment # 11: Line 224-255. Authors have not explained the limit of detection of the assay, sensitivity and specificity with reference to gold standard.

Response: We are so grateful to you for raising this question. Accordingly, we used gold-labeled indirect immunofluorescence to verify the specificity and sensitivity of our developed method and compared the limit sensitivity of the method. We have added this section to Figure S1 and Figure S2-S12 in the Supplementary Material.

Finally, we would like to express our sincere gratitude to the editors and reviewers' for considering our manuscript for publication.

I look forward to hearing from you.

Sincerely,

Meilin Jin

Wuhan, 430030, P. R. China

Tel: 027-87286905

Fax: 027-87286900

E-address: jml8328@126.com

September 24, 2022

Prof. Meilin Jin
State Key Laboratory of Agricultural Microbiology, College of Veterinary Medicine, Huazhong Agricultural University
Wuhan
China

Re: Spectrum02343-22R1 (One-step rapid and sensitive ASFV p30 antibody detection via Nanoplasmonic biosensors)

Dear Prof. Meilin Jin:

I am glad to inform you that your manuscript has been accepted, and I am forwarding it to the ASM Journals Department for publication. You will be notified when your proofs are ready to be viewed.

Sincerely,

Biao He
Editor, Microbiology Spectrum
